# In Vitro Expression Analysis Reveals HML6-c14 to Be an Attractive Research Target

**DOI:** 10.3390/biom13091378

**Published:** 2023-09-12

**Authors:** Takaya Oda

**Affiliations:** Department of Human Molecular Genomics, Faculty of Medicine, University of the Ryukyus, Uehara 207, Nishihara, Nakagami 9030215, Okinawa, Japan; toda@med.u-ryukyu.ac.jp; Tel.: +81-98-895-1201; Fax: +81-98-895-1439

**Keywords:** human endogenous retrovirus, placental expression, long non-coding RNA, nuclear retained transcript

## Abstract

HML6-c14, a long terminal repeat (LTR)-type retrotransposon identified by expressed sequence tag (EST) database screening, was found to undergo RNA processing resembling that of placental tissue by in vitro expression analysis. Previous in situ hybridization studies using normal placental tissue showed that the transcript remained in the nucleus. However, among the transcripts forcedly expressed in cultured cells, the transcript that retained the 3.3 kb intron was observed in the nucleus, and a part of the spliced transcript was observed outside the nucleus. To verify whether this cytoplasmic transcript could be translated, we examined the coding potential of the open reading frame (ORF), consisting of 109 codons on the spliced transcript, along with two other putative ORFs detected in the intronic region. As a result, none of the ORF-derived products could be detected by Western blotting as fusion proteins tagged with the FLAG epitope, suggesting that HML6-c14 belongs to a group of long non-coding RNA (lncRNA) genes. Promoter analysis of the upstream 6.4 kb genomic region also suggested that the 5′-LTR itself potentially retains high promoter activity. Despite losing the ability to produce functional proteins, HML6-c14 continues to retain its transcriptional ability while converting to an lncRNA gene, which is an interesting subject for future research.

## 1. Introduction

Due to retrovirus infection of the germline of a primate ancestor, human endogenous retroviruses (HERV) were integrated into our genome [1] and subsequently transmitted vertically to subsequent generations in a Mendelian fashion [2,3]. Because of their ongoing replication and retrotransposition, multicopy proviruses are scattered throughout the human genome, with HERVs constituting nearly 8% of our DNA [4,5]. Most of these “fossil” viruses remain inactive due to mutations that have accumulated over the course of evolution [6]. However, a small fraction of HERVs are still expressed as RNAs that are translated into functional proteins, including products that have been implicated in various normal physiological processes such as embryogenesis [7]. Recent studies have shown that the HERV most recently integrated in the human genome, HERVK (also known as HML-2 (Human endogenous mouse mammary tumor virus-like subgroup 2)), is reactivated in senescent human cells, resulting in the generation of retrovirus-like particles (RVLPs); these RVLPs are responsible for senescence in neighboring young cells. These findings indicate that the resurgence of specific HERVs may serve as a hallmark and driving force of cellular and tissue senescence [8].

On the other hand, non-coding RNA (ncRNA) is a type of RNA molecule that lacks protein-coding properties. Among ncRNA, long non-coding RNA (lncRNA) is generally defined as transcripts of 200 nucleotides or more, a size range larger than molecules such as microRNA (miRNA) and small nuclear RNA (snRNA), classes collectively referred to as small ncRNA (sncRNA) [9]. Some lncRNAs are known to undergo post-transcriptional modifications such as splicing, cap addition to the 5′ end, and polyadenylation of the 3′ end, like messenger RNAs (mRNAs). lncRNAs are increasingly recognized as key regulatory transcripts that influence both physiological and pathological processes [10]. In recent years, RNA sequencing (RNA-seq) experiments have been conducted on various human tissues and cell samples, and a large number of RNAs classified as lncRNAs have been discovered. A review of changes in the number of existing genes on the human genome, as cataloged by the GENCODE project [11], reveals that the number of protein-coding genes has remained largely constant at approximately 20,000. However, the number of lncRNA genes continues to increase year over year. The proportion of these genes for which some literature information exists is over 70% for protein-coding genes but less than 2% for lncRNA genes. The remaining 98% of lncRNA genes remain uncharacterized in the literature, and the functions of most lncRNAs have yet to be elucidated [12,13].

A search (using a typical genome portal site) of the genome interval extending from base 69,811,463 to 69,816,023 of chromosome 14 (chr14:69,811,463-69,816,023) within the human genome reference sequence (version GRCh38/hg38) reveals the presence of HML6-c14 (also known as 14q24.2 reported by Pisano et al.) [4], the target gene of the research described here. HML6-c14 is a 4561 bp long terminal repeat (LTR)-type retrotransposon that exhibits specific expression in the placenta. The presence of multiple related entries in the expressed sequence tag (EST) database maintained by the National Center for Biotechnology Information (NCBI) suggests that this element is transcribed, and HML6-c14 has been identified as a gene in several experiments [14]. Like ERVW-1 (endogenous retrovirus group W member 1, envelope), another HERV gene that has been reported to be involved in placental function [15], HML6-c14 is expressed as two types of transcripts, those with and without splicing. However, unlike ERVW-1, which has an open reading frame (ORF) encoding the syncytin-1 protein consisting of 538 codons, the HML6-c14 transcript has an ORF corresponding to at most 164 codons. On the other hand, in situ hybridization (ISH) using normal placental tissue shows that the HML6-c14 transcript is associated with the nucleus [16], suggesting that this element may generate a nuclear-localized lncRNA. Among this type of lncRNA, NEAT1 (Nuclear Enriched Abundant Transcript 1) [17] and MALAT1 (Metastasis Associated Lung Adenocarcinoma Transcript 1; also known as NEAT2) [18] have been the most intensively studied. NEAT1 is involved in the regulation of gene expression; MALAT1 is involved in transcriptional regulation, splicing, and cancer metastasis. Since ERVW-1 was first reported in 2000, there have been successive reports that other HERVs are involved in human physiology [19,20]. We therefore sought to determine whether HML6-c14 is also involved in human physiology. As a first step, we determined the characteristics of HML6-c14 using an in vitro system and examined the potential function of the lncRNA encoded by this element.

## 2. Materials and Methods

### 2.1. 5′ Rapid Amplification of cDNA Ends (RACE) Analysis

Here 5′-RACE was performed using the 5′ RACE System for Rapid Amplification of cDNA Ends, Version 2.0 (Thermo Fisher Scientific, Waltham, MA, USA), according to the manufacturer’s instructions, with slight modifications. Briefly, 5 µg of human placental total RNA (BioChain Institute Inc., Newark, CA, USA) was reverse transcribed (RTed) using the gene-specific primer (GSP) 1 (primer i in Figure 1a), and the RT products were column purified after RNase treatment. These purified single-stranded DNAs were used as a template for first polymerase chain reaction (PCR) after adding dA-tail using terminal nucleotidyl transferase (TdT). The second PCR product was gel-purified and sequenced by the Sanger method. The dT adaptor used in the first PCR and the adaptor primer used in the second PCR are shown in Table 1, along with the GSPs (primers i and j).

### 2.2. Generation of Expression Constructs

The expression construct pCMV-c14-w was generated as follows. First, the cytomegalovirus (CMV) enhancer and promoter regions of pAcGFP1-C1 were amplified by PCR with primers e and f (as indicated in Figure 1b). The resulting product was digested with AseI and XbaI and inserted into similarly digested pAcGFP1-C1. The resulting plasmid is identical to the parent except that the CMV transcription initiation site and the downstream *gfp* ORF (encoding green fluorescent protein (GFP)) were removed. Second, in a separate cloning reaction, a 4203 bp amplicon corresponding to HML6-c14 was obtained by nested PCR (Figure 1a: primers a and b for first PCR, primers c and d for second PCR) using Human RPCI-11BAC (Roswell Park Cancer Institute (Buffalo, NY, USA), Library 11, bacterial artificial chromosome) Clones 605L8 DNA. This clone was purchased from Advanced Geno Techs Co. (Tsukuba, Japan) together with clone 313N23. The resulting product was cloned into the pBluescript II (Agilent Technologies, Santa Clara, CA, USA) cloning vector, and the identity of the insert was confirmed by sequencing. Third, the HML6-c14 gene was subcloned into the modified pAcGFP1-C1 vector via XbaI digestion. The resulting plasmid (designated pCMV-c14-w) places the HML6-c14 ORF, preceded by the native HML6-c14 transcription initiation site (as defined above by 5′-RACE), between the CMV enhancer/promoter and the simian virus 40 (SV40) polyA signal. Again, the sequences of the primers used for the above procedures are listed in Table 1.

### 2.3. Cell Culture and Transfection

Cells were grown at 37 °C in a humidified 5% CO_2_ atmosphere using the following media (all supplemented with 10% fetal bovine serum): Eagle’s minimal essential medium for HeLa cells, Dulbecco’s modified Eagle’s medium for Cos7 cells, and Dulbecco’s modified Eagle’s medium/Ham’s nutrient mixture F-12 for BeWo cells. For HeLa and Cos7, the media were additionally supplemented with antibiotics (penicillin and streptomycin, 100 U/mL each). Transfection studies were performed using FuGENE^®^ 6 (Promega, Madison, WI, USA) according to the manufacturer’s instructions. Briefly, 24-well plates were seeded with cells at a density of 1.0 × 10^5^/well for 24 h before transfection. On the day of transfection, a mixture of transfection reagent and DNA was added to each well of cells to be transfected. The amounts of transfection reagent mixed with 0.5 µg of plasmid DNA for HeLa, Cos7, and BeWo were 1.5, 1.5, and 2.5 μL, respectively. After 48 h, cells were processed as indicated below.

### 2.4. Probe Preparations for Northern and ISH Experiments

Single-stranded riboprobes were prepared using the Roche DIG RNA labeling kit (SP6/T7) (Roche, Basel, Switzerland) according to the manufacturer’s instructions, with minor modifications. To prepare probes A and B (shown in Figure 1b), the corresponding 219 bp and 560 bp PCR products (respectively) were cloned into the pGEM cloning vector (Promega, Madison, WI, USA); the resulting plasmids were used as labeling templates after sequence confirmation. The labeling template for generating probe C was constructed using the primers and vectors reported previously [16]. For the *gfp* probe, a 1061 bp amplicon containing the GFP-encoding ORF was obtained by PCR using primers GFP probe sense and GFP probe antisense and plasmid pAcGFP1-C1 as the template; this PCR product was digested with BsaHI/BglII, and a 713 bp *gfp* DNA fragment was inserted into the pBluescript II cloning vector. The *gfp* probe itself was prepared using the same kit as above, except that the nucleotide triphosphate (NTP) mixture containing digoxigenin-11-uridine triphosphate (DIG-11-UTP) was replaced with an NTP mixture containing fluorescein-12-UTP manufactured by the same company. Again, the primer sequences used for the above procedures are listed in Table 1.

### 2.5. Real-Time PCR Analysis

Total RNA was extracted using TRIzol^®^ reagent (Invitrogen, Waltham, MA, USA) according to the manufacturer’s instructions. An aliquot (100 µg) of the resulting total RNA was polyA purified using Oligotex™-dT30<Super> (Takara Bio Inc., Shiga, Japan), again according to the manufacturer’s instructions. An aliquot (2 μL) of polyA equivalent to 0.5 ng RNA was then diluted into 6 mL of diethyl pyrocarbonate (DEPC)-treated water and used as the template in an RT reaction using TaqMan™ Reverse Transcription Reagents (Applied Biosystems, Waltham, MA, USA) according to the manufacturer’s protocol. The resulting products were employed for real-time quantitative PCR using the Power SYBR™ Green PCR Master Mix (Applied Biosystems, Waltham, MA, USA) in the StepOnePlus™ (Applied Biosystems, Waltham, MA, USA) system. Target RNA expression levels were estimated using the relative standard curve method. The standard curve was generated using serial dilutions of pAcGFP1-C1-based plasmids containing the respective target sequences. Again, the primers realtime_GFP sense and realtime_GFP antisense used in this study are listed in Table 1.

### 2.6. Northern Analysis

HeLa cells were co-transfected with pCMV-c14-w and pAcGFP1-C1 (Clontech Laboratories, Inc., Mountain View, CA, USA) at a 1:1 (*w*/*w*) ratio. After 48 h, total RNA was extracted from the cells and polyA purified as described above. A portion of the polyA RNA was subjected to RT, and the copy number of the *gfp* transcript in the original polyA RNA sample was estimated by real-time PCR. Two sets of polyA RNA samples containing 1.0 × 10^9^ copies of the *gfp* transcript were separated on a denaturing 1.0% formaldehyde agarose gel using separate lanes along with an RNA size marker. Following electrophoresis, a gel fragment of the same size as the marker was excised. A portion of this gel fragment was stained with ethidium bromide, then photographed under ultraviolet (UV) light. The remaining gel fragment was transferred to a Hybond-N+ nylon membrane (Amersham Biosciences Corp., Piscataway, NJ, USA). Another blot was made in the same manner, and each blot was bisected so that each contained an RNA sample lane.

Hybridization was performed using Clontech’s ExpressHyb^™^ (Clontech Laboratories, Inc., Mountain View, CA, USA) according to the manufacturer’s protocol. Briefly, the resulting blots were prehybridized at 68 °C for 30 min in 5 mL of ExpressHyb solution, and then the ExpressHyb Solution was replaced with the same volume of fresh solution containing the DIG-labeled probes for HML6-c14 (probes A, B, and C). Hybridization was performed for each probe using one blot prepared by the method mentioned above. After incubating for 1 h at 68 °C, each blot was subjected to stringent washes. The signal from each probe was detected using the DIG Luminescent Detection Kit (Roche, Basel, Switzerland) and the ImageQuant™ LAS 4000 mini chemiluminescence imaging system (GE Healthcare, Chicago, IL, USA) according to the respective manufacturer’s instructions. After acquiring the DIG-labeled probe image, the membrane was stripped, blocked, and incubated overnight with an anti-fluorescein antibody (Roche, Basel, Switzerland). The blot was then exposed to peroxidase-conjugated goat anti-mouse immunoglobulin G (IgG; H+L) antibody (Jackson Immuno Research Laboratories, Inc., West Grove, PA, USA) as a secondary antibody, and the signal from the fluorescein-labeled probe was detected with ECL-prime (GE Healthcare, Chicago, IL, USA) and the chemiluminescence imaging system as described above.

### 2.7. In Situ Hybridization (ISH)

HeLa cells were grown on 24 mm × 24 mm glass coverslips submerged in 35 mm diameter culture dishes, then transfected with pCMV-c14-w. After 48 h, coverslips were washed with phosphate-buffered saline (PBS) and fixed with 4% paraformaldehyde (PFA) in PBS for 10 min on ice. After rinsing in PBS, cells were permeabilized with 0.2% Triton X-100 in PBS for 10 min on ice and then washed three times with PBS before being processed for ISH. ISH was performed as previously described [16]. Briefly, the coverslips were washed in PBS (5 min × 2), treated with 4 µg/mL Proteinase K (Takara Bio Inc., Shiga, Japan) in 20 mM Tris-HCl (pH 8.0) and 1 mM EDTA at 37 °C for 1.5 min, washed with PBS (5 min), and re-fixed in 4% PFA in PBS for 5 min. The coverslips were then incubated at room temperature in 0.1 M triethanolamine plus 0.25% acetic anhydride for 10 min. Following acetylation, the coverslips were washed with PBS and saline (5 min each), dehydrated in an ethanol dilution series (30%, 50%, 70%, 90%, 100%; 2 min each), and dried at room temperature.

Anti-sense or sense complementary RNA (cRNA) probes were denatured and suspended in a hybridization buffer (In Situ Hybridization Reagent 7 (ISHR 7); Nippon Gene CO., LTD., Toyama, Japan). The riboprobe mixtures were placed on individual 24 mm × 24 mm coverslips; following air drying, larger 24 mm × 40 mm coverslips were placed on the hybridization mixture with the cell grown-side down. The stacked coverslips were set in a humid container and incubated overnight at 65 °C.

After hybridization, the stacked coverslips were washed with Wash Buffer (2× SSC (0.3 M sodium chloride, 30 mM trisodium citrate, pH 7.0), 50% formamide, and 20 mM mercaptoethanol) at 37 °C for 30 min. Coverslips with cells were washed twice (65 °C, 30 min per wash) with fresh wash buffer. Next, the coverslips were washed with NTE buffer (0.5 M NaCl, 10 mM Tris-HCl (pH 8.0), and 5 mM EDTA) at 37 °C for 15 min. RNase A (20 µg/mL in NTE buffer) treatment was carried out at 37 °C for 30 min to minimize nonspecific binding, and the coverslips were washed once with fresh NTE buffer at 37 °C for 15 min. Coverslips were then washed with fresh wash buffer at 37 °C for 30 min, followed by washes with 2× SSC and 0.1× SSC at room temperature for 15 min each.

We used an HNPP (2-hydroxy-3-naphtoic acid-2′-phenylanilide phosphate) fluorescent detection set (Roche, Basel, Switzerland) to detect DIG-labeled riboprobes. The coverslips were dipped in TBST buffer (150 mM NaCl, 100 mM Tris-HCl (pH 7.5), 0.1% Tween 20) for 5 min and immersed in blocking solution (1.5% in TBST) (Roche, Basel, Switzerland) at room temperature for 2–3 h. After blocking, the coverslips were washed with TBST at room temperature for 5 min, covered with alkaline phosphatase-labeled goat anti-DIG solution (Roche, Basel, Switzerland; antibody stock solution was diluted 1:1000 in TBST), and incubated at 4 °C overnight.

Following incubation, coverslips were washed twice with TBST buffer at room temperature for 10 min and then three times (room temperature, 10 min per wash) with Buffer 3 (150 mM NaCl, 100 mM Tris (pH 7.5), 0.05% Tween 20). A mixture of 10 µL each of HNPP and Fast Red TR solution (Roche, Basel, Switzerland) in 1 mL of Buffer 4 (100 mM NaCl, 10 mM MgCl2, 100 mM Tris (pH 8.0)) was filtered through a 0.2 µm nylon syringe filter, then applied to the coverslip, which was incubated for 30 min at room temperature. After this incubation, the coverslips were washed with deionized water (dH_2_O) for 10 min to quench the chromogenic reaction. The coverslips were counterstained with Hoechst 33342 (Dojindo Laboratories, Kumamoto, Japan) solution (1:100 in PBS), and the signals were observed under a fluorescence microscope (BX51; Olympus, Tokyo, Japan). The fluorescence observation settings were excitation at 330–385 nm and emission at 420 nm < for Hoechst 33342 and excitation at 520–550 nm and emission at 580 nm < for HNPP.

### 2.8. Analysis for Protein-Coding Potential

The SeqBuilder Pro module of Lasergene Molecular Biology (DNASTAR, Inc., Madison, WI, USA) was used for ORF analysis. PCR was performed to obtain the target DNA fragments, which were individually cloned into the pBluescript II cloning vector. For the clone of each PCR-amplified fragment, the identity of the insert was confirmed by sequencing, and a separately prepared linker encoding a FLAG tag (peptide sequence DYKDDDDK) was inserted in-frame at the downstream end of each ORF. Subsequently, the entire insert region (containing both the ORF and the FLAG tag-encoding sequence) was used to replace the *gfp* ORF region of pAcGFP1-C1, yielding an expression construct encoding the respective FLAG-tagged protein. Two types of positive control constructs were prepared. The first, the hRluc (humanized (codon-optimized for mammalian expression) *Renilla* luciferase) control construct, was generated by using pGL4.74 (Promega, Madison, WI, USA) as the template for PCR and cloning, yielding an expression plasmid encoding *Renilla* luciferase with a C-terminal FLAG tag. The second, the GFP control construct, was generated by directly inserting another FLAG tag-encoding linker into pAcGFP1-C1, yielding an expression plasmid encoding GFP with a C-terminal FLAG tag. The primers and linkers used in the above experiments are listed in Table 2.

HeLa cells were transfected with each construct; after 48 h of growth, cell extracts were prepared in 100 µL of Cell Lysis Buffer (50 mM Tris-HCl (pH 7.4), 150 mM NaCl, 1 mM EDTA, 1% Triton X-100, supplemented with protease inhibitor cocktail (Roche, Basel, Switzerland)). An aliquot (7.5 μL) of each cell lysate was separated by standard sodium dodecyl sulfate-polyacrylamide gel electrophoresis (SDS–PAGE), transferred to a membrane, and analyzed by probing with monoclonal anti-FLAG M2 antibody (Sigma-Aldrich, St. Louis, MO, USA; primary antibody), followed by peroxidase-conjugated goat anti-mouse IgG (H+L) antibody (Jackson Immuno Research Laboratories, Inc, West Grove, PA, USA; secondary antibody). To check the loaded amount of cell lysate based on the amount of β-actin, a monoclonal anti-β-actin antibody (FUJIFILM, Tokyo, Japan; primary antibody) and a peroxidase-conjugated goat anti-mouse IgG (H+L) antibody (Jackson Immuno Research Laboratories, Inc., West Grove, PA, USA; secondary antibody) were used after stripping the anti-FLAG antibody. Antibody binding was detected using ECL-prime, and the chemiluminescence signal was detected by exposing it to an X-ray film (Fuji medical X-ray film Super RX; FUJIFILM, Tokyo, Japan) and developing and fixing it manually using GBX Chemicals (Carestream, Rochester, NY, USA). The blot was superimposed on the film to confirm the position of the size marker before acquiring an image with the scanner (MG8130; Canon Inc., Tokyo, Japan) for analysis.

### 2.9. Promoter Analysis Targeting the 6.4 kb Upstream Flanking Region of HML6-c14

#### 2.9.1. Generation of Luciferase-Encoding Constructs

Human RPCI-11BAC clones 605L8 and 313N23 were used as PCR templates or restriction enzyme digestion substrates. PCR products amplified using BAC DNA as a template were cloned into the pBluescript II cloning vector. The DNA fragments corresponding to bp 81,780–82,205, 81,332–82,205, and 80,581–82,205 of BAC clone 605L8 on Chromosome 14 were then subcloned from the pBluescript construct into the multiple cloning site (MCS) of pGL4.10 (Promega, Madison, WI, USA) using the respective restriction enzymes. Sequences corresponding to bp 75,401–78,764 and 78,764–81,332 of chromosome 14 (i.e., regions upstream of the HML6-c14) were excised from the BAC clones (using either the combination of KpnI and BamHI or BamHI alone, respectively) and (separately) inserted upstream of the bp 81780–82205 sequence in the corresponding pGL4.10 construct. For comparison with these HML6-c14 plasmids, three constructs for ERVW-1 were also prepared as described by Yu et al. [21]. Briefly, the 2072 bp DNA region containing the entire bp 19,661–21,360 of BAC clone 313N23 on chromosome 7 was amplified by PCR and cloned into a vector. Next, DNA fragments corresponding to bp 19,661–21,360, 20,881–21,360, and 21,132–21,360 were subcloned (separately) into the MCS of pGL4.10 via appropriate restriction enzyme sites (using sites within the amplicon or within the MCS of the cloning vector). The primer sequences and restriction enzymes used in the above experiments are summarized in Table 3.

#### 2.9.2. Dual-Luciferase Reporter (DLR) Assay

All procedures were performed using the Dual-Luciferase^®^ Reporter Assay System (Promega, Madison, WI, USA) according to the manufacturer’s protocol. Briefly, BeWo cells were transfected with a mixture of two plasmids encoding either firefly luciferase (to determine the expression of the experimental constructs) or *Renilla* luciferase (as an internal control). The mixing ratio of test and internal control plasmids was 20:1 for HML6-c14 constructs and 10:1 for ERVW-1 constructs. Following lysis of cultures in 100 µL of passive lysis buffer, aliquots of 20 µL of the lysate were introduced into a TD-20/20 Luminometer (Turner BioSystems, San Jose, CA, USA), and the activities of firefly and *Renilla* luciferases were measured. The constructs were assayed in triplicate (technical replicates) as either three (HML6-c14) or two (ERVW-1) biological replicates.

### 2.10. Statistics

DLR assay results were analyzed using two-tailed non-paired Student’s *t*-tests in StatView 5 (SAS Institute, Inc., Cary, NC, USA). The *p*-values of less than 0.05 were considered statistically significant.

## 3. Results

### 3.1. 5′ Rapid Amplification of cDNA Ends (RACE) Analysis

As shown in Figure 2a, a single band was obtained by electrophoresis of the second PCR product. Sequencing of this PCR product revealed that the sequence downstream of the dT adapter used in the first PCR was identical to a part of the 5′-LTR sequence. The starting point for sequence matching was the 299th Guanosine from the 5′ end of the 5′-LTR, which was determined to be the transcription initiation site as shown in Figure 2b.

**Figure 1 biomolecules-13-01378-f001:**
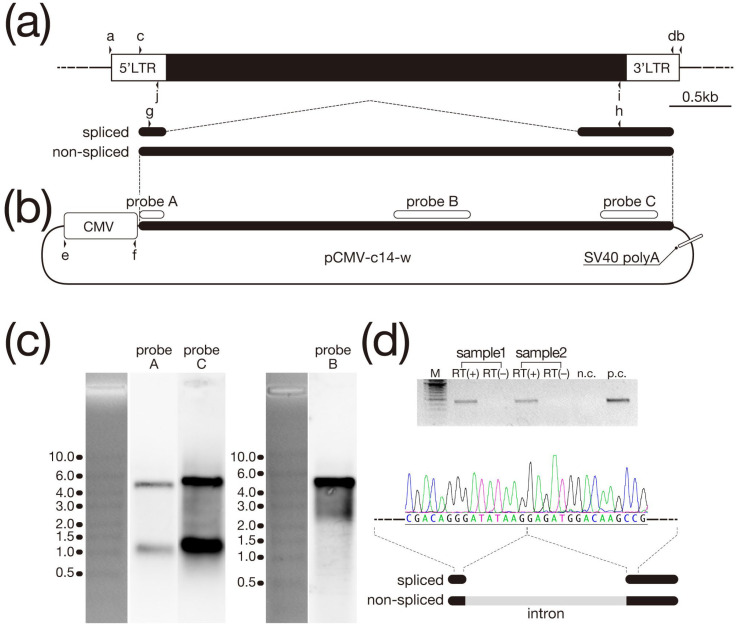
Genomic structure and Northern analysis of HML6-c14. (**a**) The genomic region spanning HML6-c14 is indicated by a black rectangle flanked by two white rectangles representing LTRs. (top) The two types of transcripts generated from the HML6-c14 element are indicated by black lines. (bottom) The eight arrowheads indicate the positions of the primers used in this experiment. (**b**) The HML6-c14 expression construct, which is derived from pAcGFP1-C1 (Clontech Laboratories, Inc., Mountain View, CA, USA), is shown. The three white lines drawn along the region where the unspliced transcript was inserted indicate the positions of the probes used in Northern analysis. Arrowheads indicate the positions of the primers used in the experiment. (**c**) The results of Northern analysis of transcripts with forced expression in HeLa cells are shown (The original image can be found at Appendix A). Images of the signals detected by the three types of probes and the positions of size markers separated by electrophoresis when the blots were prepared are shown. The unit of the size markers is kilobases. (**d**) Results of reverse transcription-polymerase chain reaction (RT-PCR) performed using primers g and h to confirm splicing. This image shows the agarose gel electrophoresis results of RT-PCR. The positive control (p.c.) represents a reaction performed using RNA extracted from placenta; the negative control (n.c.) represents a reaction performed without template RNA. The same-size band of 455 bp was detected in the two samples expressed in HeLa cells and in the positive control (top) ((The original image can be found at Appendix A)). Sequencing results of RT-PCR products recovered from the gel. Nucleotide sequences identical to the splicing pattern reported for placenta were observed (bottom) [14].

### 3.2. Northern Analysis of Transcripts Overexpressed in Cultured Cells

The results of the Northern analysis of RNA generated via forced expression of CMV-c14-w in HeLa cells are shown in Figure 1c. Two bands, of approximately 4.5 and 1.0 kb, were detected by both probes A and C using the expression construct shown in Figure 1b. On the other hand, probe B detected only the longer 4.5 kb band. Since the position of probe C within HML6-c14 is the same as that used in the original Northern analysis using placental RNA [14], we inferred that these 4.5 and 1.0 kb transcripts differ in the presence or absence (respectively) of the 3.3 kb intron that would be spliced out during RNA processing. The signal obtained with probe A is nominally weaker than that obtained with probe C; nonetheless, the results with probe A, which are almost identical to exon 1 and correspond to sequences shared between the two transcripts, support this speculation. To confirm this interpretation, RT-PCR was performed using primers g and h (shown in Figure 1a); as shown in Figure 1d, the splice junction was found to be identical to that previously reported in the placenta [14]. Since the densities of the large and small bands did not appear to differ for blots hybridized with probes A and C, we inferred that the proportions of the two transcripts produced by the presence or absence of splicing did not differ greatly from those observed in the placenta [14]. Next, we employed ISH to investigate in detail the intracellular localization of these two types of transcripts.

### 3.3. ISH analysis of Transcripts Overexpressed in Cultured Cells

The results of ISH analysis of transcripts produced in HeLa by forced expression of pCMV-c14-w are shown in Figure 3. The signal observed with probe B, which detected only the longer band (~4.5 kb) in the Northern analysis, appeared to be in perfect agreement with the nuclear signal obtained by Hoechst 33342 staining (Figure 3b). This result is very similar to the previously reported results of ISH performed on normal placental tissue using the probe, which is identical to probe C in Figure 1b [16]. However, hybridization using probe A (which, in Northern analysis, detected two bands inferred to reflect the presence or absence of splicing; Figure 1c) showed a signal in a different location than that obtained with probe B (Figure 3a). The most prominent difference was that the signal obtained with probe A was seen not only in the nuclear region (as with probe B) but also in the cytoplasm. Based on the results of the Northern and ISH analyses, we hypothesized that all or part of the spliced transcript is present in the cytoplasm. In this experiment, we used HeLa cells, which are excellent at handling and signal observation. However, since HML6-c14 is naturally expressed in placental tissue, the same result in placenta-derived cell lines such as BeWo cells would further strengthen this hypothesis. Unfortunately, our attempts at preparing splice-type transcript-specific probes have been unsuccessful so far.

Sequence analysis of the spliced transcripts detected the presence of only one ORF of greater than 100 codons within Exon 2. Next, we examined the possibility that this ORF on the spliced transcript, which appeared to accumulate in the cytoplasm, is translated along with the other two ORFs present within the intron, as summarized in Figure 4a.

### 3.4. Investigation of Translation Potential by Western Blot

The ORF search results for HML6-c14 are summarized in Figure 4a. ORFs comprising more than 100 codons were found at bp +1357 to +1851 (ORF1: 164 codons), +1854 to +2195 (ORF2: 113 codons), and +3777 to +4106 (ORF3: 109 codons) within the unspliced transcript. Both ORF1 and ORF2 are predicted to be present only in the unspliced transcript. When each ORF was searched by BLASTP on the NCBI site, the following entries were displayed at the top by scoring. Note that entries that are thought to be derived from the HML6-c14 locus have been intentionally excluded. ORF1 showed 62/167 (37%) identities with XP_055227821.1, endogenous retrovirus group K member 113 Gag polyprotein-like (Gorilla gorilla gorilla), ORF2 showed 75/100 (75%) identities with XP_055101502.1, deoxyuridine 5′-triphosphate nucleotidohydrolase-like (Symphalangus syndactylus), and ORF3 showed 24/34 (71%) identities with XP_032024861.1, uncharacterized protein LOC116478081 (Hylobates moloch). HeLa cells were transfected with three expression constructs (encoding each of these three ORFs as FLAG epitope-tagged proteins) and two positive control constructs, as shown schematically in Figure 4b; at 48 h post-transfection, cell lysates were prepared and subjected to Western blot analysis (Figure 4c). Signals from the fusion proteins (GFP-FLAG and hRluc-FLAG) were observed at the expected positions in both of the control cultures. However, no signal was detected from any of the tested ORF constructs, indicating that it is unlikely that any of these three HML6-c14 ORFs with 100 or more codons retain the ability to be translated into proteins. Therefore, we concluded that transcripts generated from HML6-c14 belong to the lncRNA class. Of course, in order to make this conclusion more certain, it is necessary to consider the possibility that the translated protein was unstable and could not be detected by the current method. Next, we investigated the transcription of this gene, despite its lack of translation.

### 3.5. Promoter Analysis of the 6.4 kb HML6-c14 Upstream Flanking Region

Figure 5c shows the results of the promoter analysis. The vertical axis of the graph shows the relative luciferase activity obtained by normalizing firefly luciferase activity to that of the internal control (*Renilla* luciferase). A comparison of the five tested constructs revealed that the construct incorporating sequences spanning bp 80581–82205 (containing the 1524 bp located immediately upstream of the HML6-c14 transcription initiation site) had significantly higher activity than any of the other four constructs. On the other hand, no significant difference was detected among the remaining four constructs. The numbers below the graph indicate the comparative relative luciferase activities when normalized to those obtained with the parent plasmid (pGL4.10, for which activity was defined as 1). The construct incorporating HML6-c14 sequences spanning bp 81780–82205, which closely matches the 5′-LTR region, provided a 921-fold increase in luciferase activity. In contrast, the construct containing ERVW-1 upstream sequences, including those spanning ERVW-1 bp 20881–21360 (containing a 123-nucleotide upstream flanking region), provided luciferase activity only approximately 150-fold that of pGL4.10 (Appendix A) [22].

## 4. Discussion

HML6-c14, which was first reported in 2004 [14]. As of 2008, the transcript of this element was shown (by ISH) to be localized in the nucleus of normal placental cells, suggesting that HML6-c14 is expressed as a lncRNA [16]. In the present study, which conducted a more detailed examination of this transcript using cultured cells (as shown in Figure 1c), two types of transcripts were identified from the HML6-c14 genomic region; the two transcripts appear to reflect the presence or absence of an intron, consistent with the original report examining transcripts in placental tissue [14]. Evaluation of the nucleotide sequence of the spliced transcript indicated a junction between the two exons that was identical to that of the placenta-derived transcript (Figure 1d). Furthermore, as shown by ISH (Figure 3), the unspliced transcript appears to be retained in the nucleus. This finding is consistent with the results reported for normal placental tissue in 2008 [16]. On the other hand, experiments using probes that could detect both types of transcripts revealed new insights. As shown in Figure 3a, the ISH signal extends beyond the nuclear region into the cytoplasm. This result strongly suggests that the spliced transcript also accumulates in the cytoplasm. Unfortunately, our attempts to design probes that specifically detect only spliced transcripts were unsuccessful, precluding clarification of the extent or mechanism of nuclear retention of the splice transcripts. However, even if the design of splice-form transcript-specific probes fails, it might be possible to clearly obtain nuclear and cytoplasmic fractions from placenta-derived cell lines such as BeWo using detergents. Copy number counts in each fraction might then provide experimental results that complement the results of ISH in placental tissue, where HML6-c14 is naturally expressed. Indeed, it is widely known that mRNA processing, including splicing, is linked to nuclear export [23]. Thus, the observation of the spliced transcript in the cytoplasm in this expression analysis (using cultured cells) is not unexpected. Comparing the results of our experiments to those of the previous ISH analysis (performed using placental tissue) indicates that the former is clearly superior in terms of resolution, including the ability to define the nucleus by nuclear staining. In other words, apart from quantitative issues, it is difficult to definitively exclude the possibility that HML6-c14 transcripts may also be present in the cytoplasm of placental tissue cells. Considered together, these results indicate the HML6-c14 transcript may not exist solely as a nuclear product (i.e., a nuclear lncRNA).

Clearly, ORFs consisting of more than 100 codons in HML6-c14 transcripts that remain in the nucleus are not expected to be translated, as confirmed in the present study [24,25]. However, for ORFs shorter than that, we cannot discuss the possibility of protein translation until all analysis is complete. Nonetheless, it remains possible that the HML6-c14 transcripts detected in the cytoplasm, a class that includes the spliced HML6-c14 transcript, are translated into proteins. When an ORF search was conducted to evaluate this possibility, we found that Exon 2, which is retained in the spliced transcript, includes an ORF of 109 codons (Figure 4a). Therefore, the constructs shown in Figure 4b were prepared to verify whether a total of three regions, including the two putative ORFs detected in regions other than Exon 2, retained the ability to be expressed as proteins. A total of five constructs, including the two positive control plasmids encoding GFP and hRluc, were designed to provide a FLAG epitope tag at the C-terminus of each predicted protein, thereby enabling detection of the resulting fusion proteins if the target sequences were translated. Following transfection of HeLa cells with each construct, cell lysates were subjected to Western blotting using an anti-FLAG antibody (Figure 4c). A single band corresponding to the expected molecular weight was detected in both positive control lanes, but none were detected in the three lanes corresponding to the putative HML6-c14 ORFs. Thus, we inferred that none of the HML6-c14-derived transcripts harbor functional ORFs, indicating that these transcripts are lncRNAs regardless of their subcellular localization. These results beg the question of what role is served by HML6-c14 transcripts, either in the cytoplasm or nucleus. Recent reports suggest that splicing-defective mRNAs whose introns have not been removed or have not yet been spliced are retained in nuclear structures called nuclear speckles [24,25,26]. As far as we can see in Figure 3a,b, the signal observed in the nucleus is relatively uniform, despite the presence of some densities, and does not appear to be concentrated in nuclear speckles, suggesting that an unknown mechanism might be responsible.

In Figure 1 and Figure 3, HML6-c14 was overexpressed in HeLa cells, and Northern analysis, or ISH, was performed. However, when compared with control cells expressing GFP (to monitor transfection efficiency), no obvious changes were observed under a microscope. In other words, the forced expression of HML6-c14 in HeLa cells had no adverse effects detectable under normal light microscopy. Similar results were obtained upon forced expression in Cos7 and BeWo, two cell lines derived from other tissues. HML6-c14 is originally expressed in the placenta, and BeWo cells are one of the placenta-derived cell lines. Therefore, in terms of silencing, if either transcript could be silenced separately or together in BeWo cells by techniques such as shRNA, it might have provided some clues about the biological significance of HML6-c14. To further characterize HML6-c14, we performed promoter analysis of the 6.4 kb interval upstream of the transcribed region, an interval that includes the 5′-LTR (Figure 5). This work was intended as a first step toward conducting loss-of-function experiments via genome editing. Appendix A shows the results of promoter analysis of ERVW-1, which was performed as a reproduction of the experiments of Yu et al. [21] and to confirm the sensitivity of our DLR assay; our results matched those of the previous report. The gray vertical line indicated by arrowheads in the target region of the analysis in Appendix A shows the position of a recognition sequence ((A/G)CCC(T/G)CAT) for the transcription factor GCM1 (glial cells missing-1) [21,27,28]. When examining the difference in expression between constructs corresponding to ERVW-1 sequences bp 21132–21360 and 20881–21360, as reported by Yu et al. [21], a clear statistical significance (*p* < 0.0001) was observed. On the other hand, the construct corresponding to HML6-c14 sequences bp 80581–82205, which includes sequences up to −1524 with respect to the transcription start site of HML6-c14, had the highest mean activity among the 5 tested constructs. Statistical analysis confirmed that the activity obtained with this construct was significantly higher than that obtained with the other four constructs (*p* < 0.05 in all cases). Yu et al. suggested that both GCM1 binding sites (located at bp −2706 to −2699 and −218 to −211 with respect to the ERVW-1 transcription start site) affect transcription of the ERVW-1 element [21]. However, the recognition sequence at ERVW-1 bp −2706 to −2699 is TCCCTCAT, representing a single-nucleotide mismatch compared to the consensus GCM1 recognition sequence. A search of the upstream sequence of HML6-c14 (as analyzed in the present study) for candidate GCM1 recognition sequences, including the atypical one identified at ERVW-1, revealed the presence of a single-nucleotide mismatch (TCCCTCAT) at bp −1087 to −1080 with respect to the HML6-c14 transcription start site (as indicated by the gray vertical line with an arrowhead in Figure 5b). However, we postulate that the effect of this possible GCM1 binding site will be very limited compared to the binding site at bp −218 to −211 with respect to the ERVW-1 transcription start site.

Compared to pGL4.10 (used as the negative control in this experiment), the construct corresponding to HML6-c14 bp 81780–82205, which incorporates the 5’-LTR region (as indicated at the bottom of the graph), had 921-fold higher activity. In the case of ERVW-1, even the construct corresponding to bp 20881–21360, which contains a GCM1 recognition sequence, provided expression that was only approximately 150 times higher than that of the negative control. Thus, despite the fact that the HML6-c14 transcript has lost the ability to be translated into protein (a loss that presumably occurred following invasion of the human genome), the promoter activity of the 5′-LTR of HML6-c14 appears to have been retained. A search of the HML6-c14 region using the University of California, Santa Cruz genome portal sites (UCSC Genome Browser; https://genome.ucsc.edu/ (accessed on 5 July 2023)) suggested that the 5′-LTR can be classified as an LTR3A sequence. According to the information obtained from the RepeatMasker Track displayed on Genome Browser, the HML6-c14 5′-LTR differs from the consensus LTR3A sequence, with changes including 8.6% divergence, 5.8% deletions, and 0.2% insertions. We hypothesize that these sequence differences may be sufficient to permit the design of guide RNAs for use in genome editing intended to excise the HML6-c14 element, permitting assessment of the effects of the loss of function of this HERV.

As mentioned above, although the number of reports on lncRNA genes has increased dramatically, the literature on these elements remains limited, and their functional analysis is even more limited [13]. The present study provided some basic information on the expression of the HML6-c14 gene as a lncRNA. This work is also expected to serve as a scaffold for future functional analysis of this element. According to Kapusta et al., 75% of the lncRNAs reported in humans contain segments derived from transposable element (TE) sequences, including HERVs [29]. Notably, HML6-c14 is expressed in the placenta, an organ that develops for a limited period of time, and so this element is expected to be an excellent candidate for investigating the tissue-specific biological function of TE-derived lncRNAs. During the preparation of this manuscript, the expression of HML6-c14 was again confirmed for human ESTs and human mRNAs from GenBank using the UCSC Genome Browser. Of the 93 entries found in total, 82 were from placental tissue or placental tissue-derived tumor tissue, and none were from germ cells. One of the remaining 11 entries was unexpectedly piR-51981, one of the piRNAs reported by Girard et al. in 2006 [30]. Its location was almost in the center of the MHM6-c14 genome and complementary to the MHM6-c14 transcript. The total length was 29 nt, of which the 28th nucleotide was different from that of the genome sequence. However, according to Reuter et al., the complementarity of bases 2 to 21 seems to be sufficient for RNA cleavage [31]. Therefore, piRNA might be involved in the regulation of HML6-c14 expression. Needless to say, although BeWo placenta-derived cells were used, their physiological activity may differ from that of normal placental cells because they are derived from tumor cells. Therefore, it is necessary to confirm our findings using other resources, such as primary cultures of placental trophoblasts, before proceeding with further research.

## Figures and Tables

**Figure 2 biomolecules-13-01378-f002:**
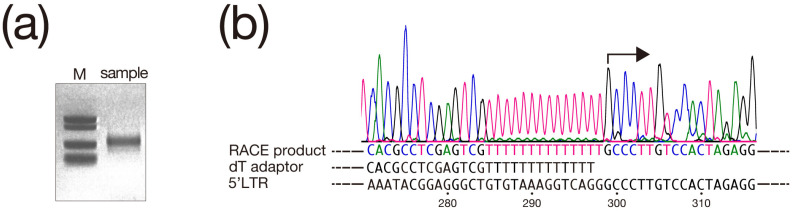
Results of 5′ Rapid Amplification of cDNA Ends (RACE) analysis, (**a**) This image shows the agarose gel electrophoresis results of second PCR using primer j in Figure 1a and adaptor primer listed in Table 1. A single band with a size between 257 bp and 521 bp compared to the size marker indicated by M is detected (The original image can be found at Appendix A). (**b**) Sequencing results of second PCR products recovered from the gel. DNA sequencing chromatogram and RACE product sequence analyzed from chromatogram are displayed. (top and second row). The sequences of the dT adaptor and 5′-long terminal repeat (LTR) are aligned according to the analysis results of the RACE products (third row and bottom). The arrow indicates the position of Guanosine, which is the transcription initiation site determined by this analysis. Numbers below the 5′-LTR sequence indicate the position of the nucleic acid from the 5′ end of the 5′-LTR.

**Figure 3 biomolecules-13-01378-f003:**
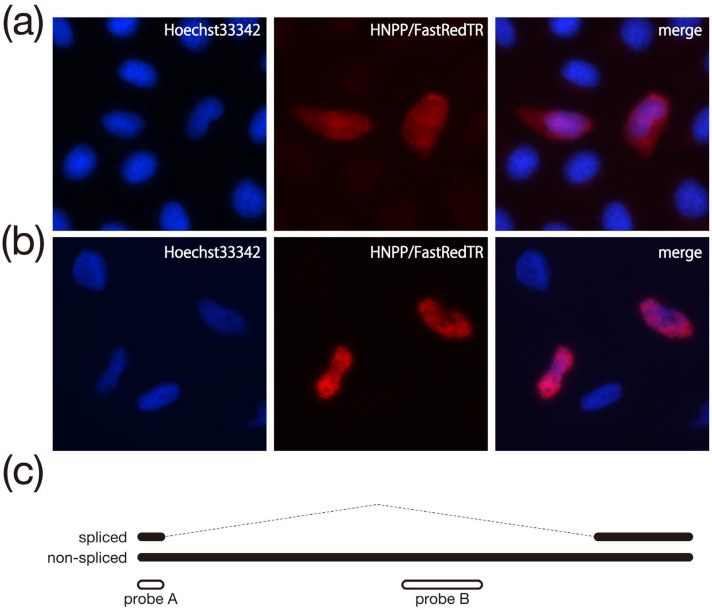
Results of in situ hybridization in HeLa cells transiently overexpressing HML6-c14. (**a**) Results using probe A. From left to right, nuclear staining with Hoechst33342, probe signal by HNPP/FastRedTR, and merged image. (**b**) Results using probe B. The type of each image and the arrangement of images are the same as in (**a**). (**c**) The two types of transcripts (with and without splicing) detected in the placenta are shown schematically by black lines (top). The white lines drawn along the black lines represent the positions of the two probes used in ISH (bottom).

**Figure 4 biomolecules-13-01378-f004:**
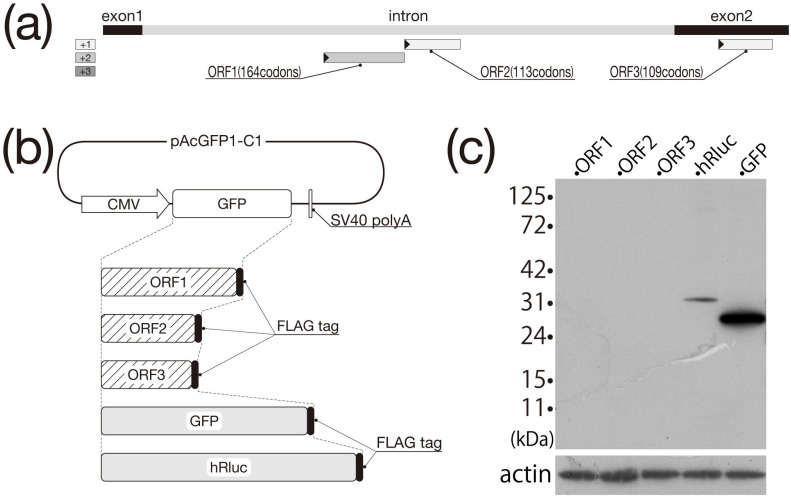
Protein translation ability analysis by detection of FLAG tag fusion proteins. (**a**) ORF search results for each frame. The shaded rectangles in each frame along the HML6-c14 genome indicate the detected ORFs. A filled triangle at the end of the rectangle indicates the position of the start codon. (**b**) Each construct prepared by modifying pAcGFP1-C1 is shown schematically. In addition to the two types of positive controls (indicated by shaded rectangles), the GFP-encoding region of the vector was replaced with a FLAG epitope-encoding linker that was ligated to the downstream end of the region to be analyzed, as indicated by the hatched rectangle. (**c**) The results of Western blotting detected with an anti-FLAG antibody are shown (The original image can be found at Appendix A). The position of the size markers is shown on the left side of the blot images. The results of probing the same blot with an anti-actin antibody (used as a loading control) are shown at the bottom of the panel.

**Figure 5 biomolecules-13-01378-f005:**
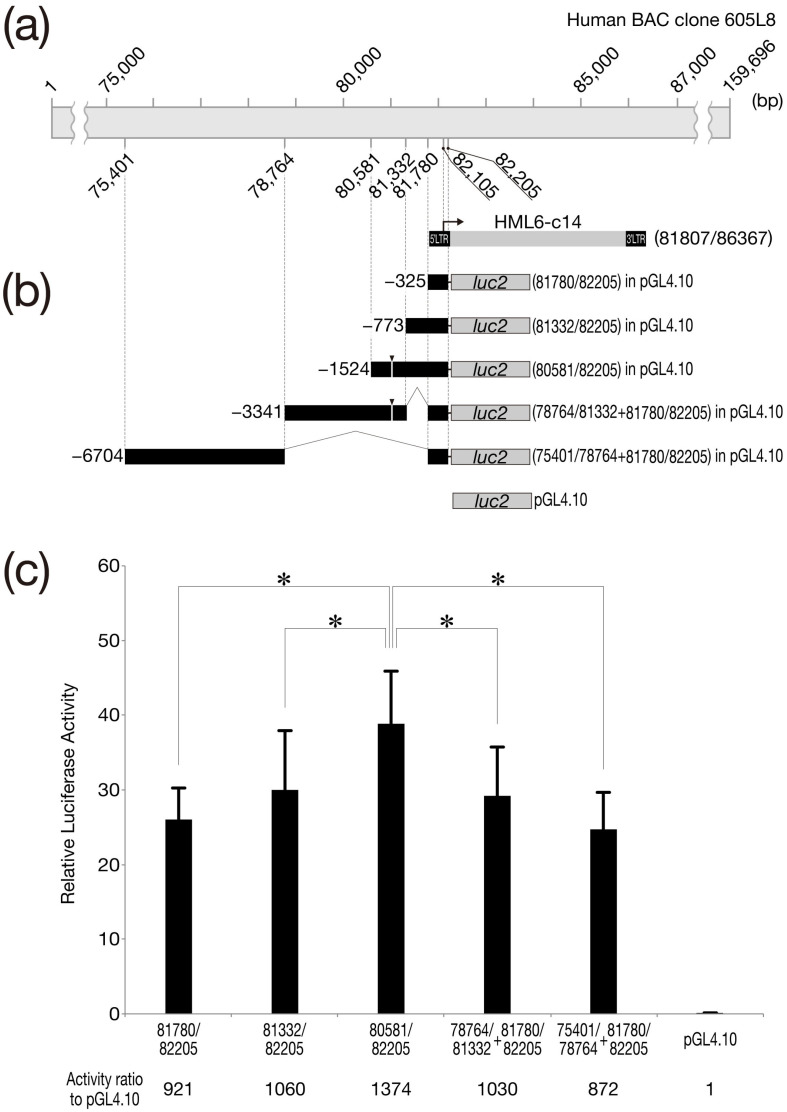
Promoter analysis of the HML6-c14 gene. (**a**) Schematic representation of human BAC clone 605L8 compared to the HML6-c14 element. The numbers above and below the 605L8 schematic denote the positions of the numbered nucleotides. The arrow above the 5′-LTR indicates the location of the transcription start site. (**b**) Schematic representation of promoter constructs used in this study. The genomic fragment inserted in pGL4.10 is indicated by a horizontal filled bar, denoting its location within BAC Clone 605L8, as depicted in Panel (**a**). The numbers in parentheses on the right indicate the ranges of the genomic fragments in BAC Clone 605L8. (**c**) Promoter analysis for the HML6-c14 gene. BeWo cells were transfected with 0.5 µg of a mixture of the indicated promoter construct and pGL4.74. Values are presented as mean and standard deviation (SD) from three independent transfection experiments, each performed in triplicate. Asterisks denote statistically significant differences (*, *p* < 0.05) between indicated expression plasmid-transfected groups (two-tailed non-paired Student’s *t*-tests). The numbers below the graph indicate the fold difference in expression following normalization compared to that obtained with the negative control (pGL4.10).

**Table 1 biomolecules-13-01378-t001:** Sequences of the oligos used in the following experiments: 5′ Rapid Amplification of cDNA Ends (RACE) analysis; Generation of expression constructs; Probe preparations for Northern and ISH experiments; and Real-time PCR analysis.

Oligo	Sequence
a	TGAGACTCCCGTGGCTCAATAGGTA
b	CCTTCCAAACCAGCCAGTTGCTT
c	AGTCTAGAGCCCTTGTCCACTAGAGGCAAG
d	ACTCTAGAGTATATTTGGAAGAAGGGTCAG
e	ATTACCGCCATGCATTAGTTA
f	ATCTAGAACTAAACCAGCTCTGCT
g	TGCATTCGCCCGCTTT
h	GTGGGCAGCTCTCATTTTCC
i	CTAGAGTATATTTGGAAGAAGGGTCAGGAAGC
j	GTCCCCGTTCACCTTCTCAAT
dT adaptor	TCGAATTGCCGACACATGAGGTTCACGCCTCGAGTCGTTTTTTTTTTTTT
adaptor primer	TCGAATTGCCGACACATGAGG
probe A sense	CTAGAGGTCAGGGCCCTTGTC
probe A antisense	GTCCCCGTTCACCTTCTCAAT
probe B sense	CAGCAGGATACATGGGACTAAT
probe B antisense	CATCTCCCATGCTGTAAGTAAGTCT
probe C sense	ACCTGGGGACAACTCAAGAAAACC
probe C antisense	GGAAGTGGAAGAGGGGGTCAGC
GFP probe sense	TTGACGCAAATGGGCGGTAG
GFP probe antisense	GAAATTTGTGATGCTATTGC
realtime_GFP sense	CGACCACTACCAGCAGAATAC
realtime_GFP antisense	CACGAAGCCGAAGTAGATCAT

**Table 2 biomolecules-13-01378-t002:** Sequences of the oligos used in the analysis for protein-coding potential.

Oligo	Sequence
ORF1_sense	CCTCATACCATGGCTGCA
ORF1_antisense	AAACGGGGTTAAGGTCTGG
ORF2_sense	GTTTTAGGCATGGGTTCC
ORF2_antisense	TGGAGAAGGGTGTAATTTGC
ORF3_sense	GGGAAAATGAGAGCTGCC
ORF3_antisense	GGCAAAAGAGATAGTGTGAAG
hRluc_sense	AAAGCCACCATGGCTTCC
hRluc_antisense	CTGCTCGTTCTTCAGCAC
linker_sense ^1^	ATCGACTACAAGGACGACGATGACAAGTAA
linker_antisense ^1^	TTACTTGTCATCGTCGTCCTTGTAGTCGAT
linker_GFP_sense	TCGACTACAAGGACGACGATGACAAGTAA
linker_GFP_antisense	GATCTTACTTGTCATCGTCGTCCTTGTAG

^1^ These were used as linkers for constructs other than GFP.

**Table 3 biomolecules-13-01378-t003:** Sequences of the oligos and restriction enzymes used in the promoter analysis targeting the 6.4 kb upstream flanking region of HML6-c14. The constructs for ERVW-1 are used for validating the sensitivity of the DLR assay.

Oligo	Sequence	Restriction Enzymes
(HML6-c14)		
81780/82205_sense	TGAGACTCCCGTGGCTCAATAGGTA	KpnI/AvrII
81780/82205_antisense	TTCTGCGGCGCCACTATGTAAC	
81332/82205_sense	TGGGGCAAACAGCAGTCTTATGGA	BamHI/AvrII
81332/82205_antisense	TTCTGCGGCGCCACTATGTAAC	
80581/82205_sense	TGGGGCAAACAGCAGTCTTATGGA	KpnI/AvrII
80581/82205_antisense	TTCTGCGGCGCCACTATGTAAC	
(ERVW-1)		
2072bp_sense	ATCCAGATGGCCTGAAGTAACTGA	SacI/XhoI for (21360/21132) PstI/XhoI for (20881/21132) XhoI/XhoI ^1^ for (19661/21132)
2072bp_antisense	CCAAGATGGTAGCAGGCCGCTTCC	

^1^ One of the XhoI sites exists in the multiple cloning site (MCS) of the cloning vector.

## Data Availability

All datasets generated for this article are included in the manuscript and/or the Appendix A.

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
