# Peer review of "In Vitro Expression Analysis Reveals HML6-c14 to Be an Attractive Research Target"

_biomolecules, 2023, doi:10.3390/biom13091378_

Round 1

Reviewer 1 Report

This manuscript has identified a transcript derived from the HML6-c14 sequence. Notably, one probe detects this transcript in the nucleus, while another probe detects it in the cytoplasm. However, despite tagging the potential open reading frames (ORFs) of the transcripts, no overexpressed proteins were found. The authors have also provided further characterization of the transcript's promoter. Nevertheless, several important questions remain to be addressed in order for the manuscript to meet the criteria of Biomolecules.

Major issues:

1.     Although the transcript is found in placenta, its location was studied in HeLa cells. It would be valuable if the author could examine the location of the transcript in placenta cell lines to ensure its relevance in the intended biological context.

2.     Understanding the biological significance of the two variants of this transcript is crucial. It is recommended that the author investigate this aspect by depleting each variant individually and together using RNAi in placenta cell lines and checking the impact of depleting each transcript on cell function.

3.     To gain further insights into the transcript's distribution, it would be beneficial if the author could perform a fractionation of the nucleus and cytoplasm, followed by checking the transcript expression via northern blotting.

Suggest to have the manuscript proofread by a native English speaker.

Reviewer 2 Report

The manuscript by Takaya Oda concerns a detailed analysis of the expression and putative coding capacity of an endogenous retrovirus element, namely HML6-c14, which contributes important knowledge to the field of lncRNA research. The experiments have been described and executed in a clear way, and the conclusions are largely supported by the results, but some parts need some explaining.

 -        According to Pisano et al., J. Vir. 2019, the HML6-c14 element, which they name 14q24.2, has full coding capacity for gag and pro next to the 5’ LTR, but not for the remainder of pol or for env, as these regions have been deleted, though not the 3’ LTR. How does this compare to the schematic depiction of the element in Fig. 4? What is meant by exon 1 and 2 there, and where are these located in the structure shown by Pisano et al?

-        In line with this, to what viral gene(s) would the ORFs 1-3 in Fig. 4 belong? Is there any homology to other proteins when using Blast?

-        As far as can be seen from Fig. 4b, the putative ORFs have been cloned directly behind the CMV promotor. Would not some protein expression be expected then? Or are any original sequences still present between the CMV-ORF sequences? Could the resulting proteins have been unstable? What protein size would have been expected? 11 and 15 kDa?

-        In section 3.5, the 5’ LTR of the HML6-c14 element is given as approx. 425 nt. This is very short for a betaretrovirus LTR. Should the LTRs not be longer? Which would explain the results given in Fig. 5c where inclusion of an upstream segment significantly increases luciferase expression?

-        There should be some discussion on the expression of the HML6-c14 element in germ line cells, as now only somatic cells have been tested. It is possible that the transcribed RNA could be a template for the generation of piRNAs in the nucleus of sperm cells/oocytes/ early embryo, or be otherwise involved in silencing of the element.

-       Line 467: it is stated here that ‘it has been confirmed that HML6-c14 transcript that remain in the nucleus are not translated’, but as far as I can see from Fig. 4, the analysis was not done using the complete transcript, but only some, putatively coding, fragments of the element? Please comment.

 Minor comments:

-          Line 15 states that the ‘coding potential of an ORF of 109 codons’ was tested, but in fact three ORFs were tested.

-          Something went wrong with the font size on pages 13-14

English language is fine, only some minor modifications are needed

Reviewer 3 Report

The author have conducted studies on human endogenous retrovirus (HERV) element HML6-c14. In his findings, he reports high promoter activity, nuclear as well as cytoplasmic localizations, and a behaviour typical of long non-coding RNA for this element. The manuscript is very well structured and written, and the English language is adequate, no major spelling/grammatical mistakes were observed.

In regards to the topic in question, the role of HERVs in cellular pathophysiology is indeed controversial, and very limited information exist about their expression behaviours, therefore, the topic covered here undoubtedly add to our knowledge in this regard, even though there is still more to be revealed regarding this HERV element.

Comments:

- In line 492, the author states that overexpression of HML6-c14 did not affect cell morphology or cell division, however, neither figure 2 or 3 indicate this finding, and I could not find support for this statement in the manuscript. What experiments/analysis methods was/were utilized to reach this conclusion?

 - While the author used adequate techniques to analyse promoter activity, cellular localization, and protein expression, I feel that the study could have benefited a lot from studying overexpression of HML6-c14 in cells utilizing proteo-transcriptomic analysis, and testing effects of expression/transfection on cellular viability/mitochondrial function..etc. 

- This is indeed is a lot of work, and many experiments utilizing different molecular biological techniques were utilized. It is very peculiar that this work was only carried out by one author. As stated in the acknowledgment section, medical students also helped during a 3 months rotations, however, such experiments surely required more time and manpower to complete. Please clarify this point.

Round 2

Reviewer 1 Report

My concerns were not addressed with experimental results. Simply adding sentences to the discussion section is insufficient to provide answers to my questions. I am opposed to publishing this work unless additional experiments are included.

Author Response

Thanks